# Determination of Formaldehyde Yields in E-Cigarette Aerosols: An Evaluation of the Efficiency of the DNPH Derivatization Method

Xiaohong C. Jin, Regina M. Ballentine, William P. Gardner, Matt S. Melvin, Yezdi B. Pithawalla, Karl A. Wagner, Karen C. Avery and Mehran Sharifi *

Center For Research & Technology, Altria Client Services LLC, 601 East Jackson Street, Richmond, VA 23219, USA; Xiaohong.Jin@altria.com (X.C.J.); Regina.M.Ballentine@altria.com (R.M.B.); William.P.Gardner@altria.com (W.P.G.); Matt.S.Melvin@altria.com (M.S.M.); Yezdi.B.Pithawalla@altria.com (Y.B.P.); Karl.A.Wagner@altria.com (K.A.W.); Karen.C.Avery@altria.com (K.C.A.)
* Correspondence: Mehran.Sharifi@altria.com; Tel.: +1-(804)-335-2062

**Abstract:** Recent reports have suggested that (1) formaldehyde levels (measured as a hydrazone derivative using the DNPH derivatization method) in Electronic Nicotine Delivery Systems (ENDS) products were underreported because formaldehyde may react with propylene glycol (PG) and glycerin (Gly) in the aerosol to form hemiacetals; (2) the equilibrium would shift from the hemiacetals to the acetals in the acidic DNPH trapping solution. In both cases, neither the hemiacetal nor the acetal would react with DNPH to form the target formaldehyde hydrazone, due to the lack of the carbonyl functional group, thus underreporting formaldehyde. These reports were studied in our laboratory. Our results showed that the aerosol generated from formaldehyde-fortified e-liquids provided a near-quantitative recovery of formaldehyde in the aerosol, suggesting that if any hemiacetal was formed in the aerosol, it would readily hydrolyze to free formaldehyde and, consequently, form formaldehyde hydrazone in the acidic DNPH trapping solution. We demonstrated that custom-synthesized Gly and PG hemiacetal adducts added to the DNPH trapping solution would readily hydrolyze to form the formaldehyde hydrazone. We demonstrated that acetals of PG and Gly present in e-liquid are almost completely transferred to the aerosol during aerosolization. The study results demonstrate that the DNPH derivatization method allows for an accurate measurement of formaldehyde in vapor products.

**Keywords:** e-cigarette; e-liquid; aerosol; 2,4-DNPH derivatization; formaldehyde; "hidden formaldehyde"; formaldehyde-containing hemiacetal/acetal adducts

## 1. Introduction

Formaldehyde (FA) is classified as a Group 1 carcinogen in humans by the International Agency for Research on Cancer (IARC) [1]. Formaldehyde is a common indoor air pollutant due to its ubiquitous use in the production of various industrial products [2]. Thus, one source of human exposure to formaldehyde is its release from household products made using formaldehyde or containing formaldehyde-releaser compounds that are placed in poorly ventilated areas [3,4]. Cigarette smoke is reported as another common source of exposure to formaldehyde, which is formed as a byproduct of the combustion process of tobacco [3]. Regulations for reporting formaldehyde yields in cigarette smoke are enacted by different regulatory authorities [5,6]. More recently, the Food and Drug Administration (FDA) cataloged a list of "Harmful and Potentially Harmful Constituents" (HPHCs) of tobacco products, which includes formaldehyde [7,8]. The FDA's Guidance to Industry regarding the submission of Premarket Tobacco Applications for Electronic Nicotine Delivery Systems (ENDS) also includes formaldehyde on the list of constituents "that would potentially cause health hazards depending on the level, absorption, or interaction with other constituents" [9].

Formaldehyde yields reported in machine-generated smoke from commercially available cigarettes vary (~10–70 µg/cigarette depending on the tobacco blend, cigarette design, and intensity of the smoking conditions [10–12]). Formaldehyde has also been reported in e-cigarette emissions [13–16]. The formation of formaldehyde in e-cigarette vapor is mainly attributed to the thermal degradation of propylene glycol (PG) and glycerol (Gly) and select flavoring agents [14–21]. Though typically at much lower levels than in tobacco smoke [22,23], a wide discrepancy in formaldehyde levels (0.5–50 µg/puff) has been reported in emissions from across commercially available e-cigarette products. The formaldehyde formation in e-cigarette aerosol is indeed related to the aerosolization efficiency of e-cigarette devices, which depends mainly on vaporizer physical and electronic design (temperature control, air flow, pressure drop, etc.), as well as the quality of materials used in manufacturing the device (heating coil element, liquid-containing cartridge, and wick) [14]. Other factors that influence the formation of formaldehyde include e-liquid components (propylene glycol, glycerol, and some flavorings), the propensity of the device to "dry-puff," thereby resulting in higher vaporization temperatures, and operating parameters of the device (voltage and puffing strength) [13–16,18,20,24–27].

For instance, a drastic increase in formaldehyde emission rate (from 0.1 to 30 µg/puff) was observed by increasing the voltage applied to a single-coil device from 3.3 to 5 V [28]. Gillman et al. reported [14] that the power intensity applied on the coil is not the sole factor affecting formaldehyde emission rates and that general device design characteristics such as coil position (top or bottom), single or dual coil-head, and coil resistance play a significant role in the formaldehyde generation process that occurs during aerosolization. The authors [14] further reported that an increase in power from 5 to 9 W in a single bottom-coil induced a drastic 70-fold increase in formaldehyde emission rate as opposed to a 6-fold increase observed using a single top-coil tank.

Due to its high reactivity, its low molecular mass, and the lack of a strong chromophore, a direct determination of formaldehyde in smoke or e-cigarette aerosol is typically achieved via a derivatization step. The conventional derivatization methodology is based on an acid-catalyzed condensation reaction between carbonyl compounds and 2,4-dinitrophenylhydrazine (2,4-DNPH). This method is described in several standardized methods, including US-EPA, NIOSH, and ISO, and has been widely used outside of nicotine products. The reaction proceeds by nucleophilic addition of the hydrazine functionality to the carbonyl compound, followed by elimination of water to form the corresponding hydrazone (Scheme 1).

**Scheme 1.** Derivatization of formaldehyde by 2,4-DNPH. The red color is used to visualize the condensation site of the methyl moiety of formaldehyde within the FA-hydrazone molecule.

The DNPH derivatization approach for the determination of formaldehyde in cigarette smoke has been developed and validated by multiple organizations, including CORESTA (Centre de Coopération pour les Recherches Scientifiques Relatives au Tabac) [29], Health Canada [30], and International Organization for Standardization (ISO) [31]. The conventional DNPH method has been widely utilized over the past decades in the tobacco industry and at independent analytical testing facilities for measuring formaldehyde yields in both conventional and electronic cigarettes.

The application of the conventional DNPH derivatization methodology for trapping and quantifying formaldehyde in e-liquids and e-cigarette aerosols presented challenges, mainly due to formaldehyde's extremely low concentration [22,23], its endogenous levels in laboratory air, and its background level in DNPH reagent [26]. In order to overcome these obstacles, modifications to the existing method for analyzing cigarette smoke with

respect to sample collection (i.e., use of DNPH-coated adsorption cartridges in lieu of impingers) and an alternative derivatization method (i.e., PFBHA) were undertaken by different laboratories using various analytical techniques (i.e., HPLC–DAD, LC–MS/MS, SPME/GC–MS, and GC–MS) [14,22,24,32–34].

Despite the widespread use of DNPH derivatization for the analysis of carbonyls in e-cigarette aerosol, in a paper published in 2017 [35], the authors theorized that the DNPH method significantly underestimates formaldehyde levels produced in e-cigarette aerosol. This theory was based on the assumption that formaldehyde-hemiacetal adducts, labeled "hidden formaldehyde," are formed in aerosol by the reversible addition of glycerol (primary hydroxyl group) and/or propylene glycol to the formaldehyde carbonyl functional group during aerosolization. The formaldehyde-hemiacetal (FA-hemiacetal) adduct(s) could then undergo an irreversible dehydration reaction catalyzed by the acidity of the DNPH trapping solution or silica sorbent (DNPH cartridge) to form two cyclic acetal isomers (Figure 1) [36]. The authors stated that the sequestrated formaldehyde portion in the form of hemiacetal (FA-hemiacetal) and/or acetal (FA-acetal) would not react with DNPH to form formaldehyde hydrozone and, thus, would not be measurable by the UV or MS detection used in the method, and therefore, the DNPH derivatization is not fit to measure total formaldehyde yields in e-cigarette aerosol, due to the inaccurate estimation of a user's exposure to formaldehyde [35]. They labeled this phenomenon as "hidden formaldehyde".

**Figure 1.** Formation of FA-glycerol hemiacetal (Gly-HA) and cyclic acetals (Gly-A). The colors are used to visualize the inclusion sites of various oxygen atoms in the reaction products.

Jensen and co-authors [36] estimated that an e-cigarette user vaping at a rate of 3 mL per day would inhale $14.4 \pm 3.3$ mg of formaldehyde per day in formaldehyde-hemiacetals and extrapolated their results to suggest an estimated increase in lifetime cancer risk by up to 15 fold higher to the risk for regular smokers. However, this study was criticized for being conducted under "unrealistic" user conditions and therefore misleading with respect to real user exposure to formaldehyde [28,37,38]. In response to the Jensen et al. study report [36], several letters were addressed to the journal editor requesting the retraction of the paper based on "fundamental flaws in the experimental and cancer risk calculations" [37]. Additional studies were conducted to replicate Jensen et al.'s findings using the same (or similar) atomizer, e-liquid, and operating conditions, which concluded that under "realistic" use conditions, formaldehyde yields in e-cigarette emissions are much lower than levels measured in cigarette smoke [28,39].

This paper describes the results from an evidence-based analytic approach to provide an objective assessment of the DNPH method performance with respect to formaldehyde quantification in e-cigarette emissions. A series of experiments were conducted to elucidate the reactivity of formaldehyde-containing acetal and hemiacetal adducts (listed in Figure 2) in the presence of an acidic DNPH derivatization solution. Additional experiments were conducted to determine whether acetals were formed during the aerosolization process or by intramolecular conversion of the hemiacetals to the cyclic acetals in acidic DNPH trapping solution. The analytical procedures used for analysis of formaldehyde, formaldehyde-containing hemiacetals (Gly$\alpha$-HA and PG$\alpha$-HA), and formaldehyde-containing acetals (Gly-A and PG-A) in e-liquid and/or DNPH trapping solution are described in the upcoming section.

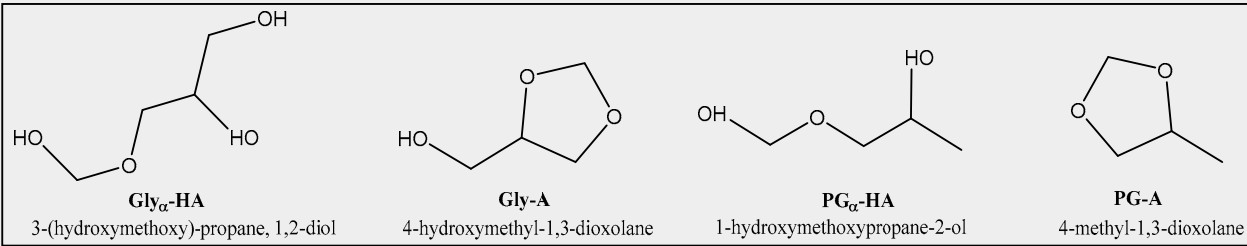

**Figure 2.** Formaldehyde-containing hemiacetal (Glyα-HA and PGα-HA) and acetal (Gly-A and PG-A) adducts.

## 2. Materials and Methods

**Test Products**. Two types of rechargeable e-cigarette devices (cig-a-like with disposable pre-filled cartridges and self-contained pod systems with refills) were purchased at retail locations in the 2018–2019 timeframe. All devices and flavors used for this study are listed in Table 1.

**Table 1.** Market test products.

| Device Type Brand ID | Flavor ID | Nicotine by Weight (%) | Product Code |
|---|---|---|---|
| Cig-a-like_A | E1 | 1.5 | CAE1 |
| Cig-a-like_B | E2 | 4.8 | CBE2 |
|  | E3 |  | CBE3 |
| Cig-a-like_C | E4 |  | CCE4 |
|  | E5 | 2.4 | CCE5 |
|  | E6 |  | CCE6 |
| Cig-a-like_D | E7 | 2.4 | CDE7 |
|  | E8 | 3.5 | CDE8 |
| Pod_E | E9 | 2.4 | PEE9 |
| Pod_F | E10 | 5.0 | PFE10 |
| Pod_G | E11 | 3.0 | PGE11 |

A reference formulation (15% water, 2.5% nicotine by weight (NBW) in a 50/50 mixture of PG and Gly) was also prepared in our laboratory in order to investigate the possible formation and transfer of formaldehyde hemiacetal and acetal adducts. Aerosols were generated using empty Cig-a-like commercial E cartridges (provided by a manufacturer) that were filled with either commercial or fortified e-liquid.

**Chemicals and Reagents.** Certified formaldehyde-DNPH hydrazone (FA-DNPH) solution in acetonitrile (700.2 μg/mL corresponding to 100 μg/mL in formaldehyde) was supplied by AccuStandard (New Haven, CT, USA). Deuterium-labeled formaldehyde-d3-3,5,6-DNPH (FA-d3-DNPH) was purchased from CDN Isotopes (Pointe-Claire, QC, Canada) and labeled as ≥99.7% pure.

The following formaldehyde-containing hemiacetal adducts, 3-(hydroxymethoxy)-propane, 1,2-diol (Glyα-HA, neat material, ≥98% pure by NMR), and 1-hydroxymethoxypropane-2-ol (PGα-HA, 50–60% pure by NMR), were custom-synthesized by Chemische Laboratorien Dr. Sönke Petersen (Worms, Germany). Glycerol formal (Gly-A) and 4-methyl-1,3-dioxolane (PG-A) were supplied by TCI (Portland, OR, USA) and Millipore Sigma (Milwaukee, WI, USA) and labeled as ≥98% pure. Certified deuterium-labeled benzene (d6-benzene) and 2,3-hexandione, used as internal standards for analysis of acetal adducts by GC–MS, were purchased from Restek (Bellefonte, PA, USA) and Alfa Aesar (Tewksbury, MA, USA), respectively.

The 2,4-dinitrophenylhydrazine hydrochloride salt (DNPH, HCl) was purchased from TCI America (Portland, OR, USA) and was labeled ≥98% pure. An acidified solution of DNPH (19 mM) was prepared in-house by dissolving purchased DNPH in acetonitrile containing 1.5% of an aqueous perchloric acid (1.82 M) solution [29]. The derivatization reagent solution was filtered and analyzed by HPLC–MS to ensure that the FA background was ≤0.05 μg/mL. A 60% solution of perchloric acid (0.6 M) was supplied by EMD

Millipore (Billerica, MA, USA). Acetonitrile and dichloromethane were distilled-in-glass grade. Type I reagent water was generated in-house as per American Society for Testing and Materials D1193 standard specification.

**Sample Generation.** E-cigarette aerosol was generated on a Borgwaldt LX20 linear smoking machine (Borgwaldt, Hamburg, Germany). The aerosol yields were obtained by collecting 50 puffs using a square-wave puff profile with a 5 s puff duration, 30 s puff interval, and a 55 mL puff volume.

The aerosol collection system for formaldehyde puffing experiments included a 44 mm-glass fiber filter pad and a 215 mm × 30 mm O.D. Drechsel-type bottle container (Prism Research Glass, Raleigh, NC, USA) enclosing the derivatization reagent (30 mL of DNPH solution). The aerosol was drawn through the filter pad followed by the impinging trap. Any formaldehyde collected on the filter pad was extracted/derivatized by adding the filter pad to the DNPH trapping solution. One milliliter of aerosol extract was then transferred to an amber autosampler vial containing 25 μL of pyridine (to stop the derivatization), and then 50 μL of FA-d3-DNPH solution (2 μg/mL) was added. The sample was then analyzed using an in-house-validated UPLC–MS detection method [40].

**FA-DNPH Determination.** The FA-DNPH content in e-liquid was determined by extracting 100 mg of the sample in 30 mL of DNPH reagent, which was left at room temperature for 5 min after mixing the reactants to allow the reactions to be completed. The reaction was stopped by adding pyridine and the sample was subject to UPLC-MS (Waters, Milford, MA, USA) analysis, as described later.

**Acetal Determination.** For the acetal puffing experiments, the aerosol was collected on a 44 mm glass fiber filter pad mounted in series with an impinging glassware containing dichloromethane (20 mL) and cooled in an ice bath (0 °C) to minimize the loss of trapping solvent. After aerosol generation, the filter pad and the impinger content were combined, and then 2 mL of type 1 water and the internal standard were added (d6-benzene or 2,3-hexandione). The mixture was vortexed for 20 min and acetal adducts were extracted by liquid-phase extraction (LPE) into the organic phase (20 mL of dichloromethane). An aliquot of dichloromethane was then analyzed by GC-MS (Agilent, Santa Clara, CA, USA), as described later.

The FA-acetal levels in the e-cigarette liquids were determined by adding the internal standard (d6-benzene or 2,3-hexandione) directly to 250 mg of the sample, which was then extracted in a type 1 water:dichloromethane mixture (2:20, *v/v*). The mixture was vortexed for 20 min and an aliquot of the organic phase containing acetal adducts was then subject to GC-MS analysis.

**Analytical Methods.** The analysis of FA-DNPH was conducted by UPLC-MS using a Waters Acquity UPLC system equipped with a binary pump, autosampler, and a TQ-S-Micro triple quadrupole mass analyzer with an electrospray ionization interface (ESI) (Waters, Milford, MA, USA). The UPLC separation was performed on a reversed-phase analytical column (Acquity UPLC BEH® C18, 2.1 × 50 mm, particle size 1.7 μm) from Waters (Milford, MA, USA) using a mixture of 10 mM of ammonium acetate/methanol (98:2 *v/v*) (mobile phase A) and a mixture of acetonitrile/1-propanol (90/10 *v/v*) (mobile phase B). The gradient program was as follows: initially constant at 65% A and 35% B for 2 min, the composition was then changed to 40% A and 60% B by a linear gradient occurring within 2 min, and then restored to the initial composition within 2.7 min and kept constant for 5 min. The flow rate was constant at 0.5 mL/min and the column temperature set to 45 °C. The ESI mass spectra for FA-DNPH and FA-d3-DNPH were acquired in negative ionization mode by monitoring their respective [M-H] molecular species (*m/z* 209 and *m/z* 212, respectively). The capillary and cone voltages were set at 0.65 kV and −32 V, respectively. The source block desolvation temperature was set to 450 °C and the source temperature was set at 150 °C. The method was validated based upon the 2005 International Conference on Harmonization (ICH) guideline "Validation of Analytical Procedures: Text and Methodology Q2(R1)" [41]. Repeatability each day was 3–12.7% of RSD for the analysis of 5 independently prepared replicate samples. Over the course

of 3 days, the method variability (intermediate precision) within samples ranged from 1.66% to 14.8% %RSD. Selected ion monitoring is a specific detection technique and no interference peaks in the samples were observed. Accuracies were 90.7–106%. The limit of quantitation (LOQ) is defined as the lowest quantifiable level of formaldehyde such that the signal-to-noise ratio (S/N) is 10. The concentration of formaldehyde in the calibration standards ranged from 0.01 to 3.8 μg/mL with $R^2$ greater than 0.995 and percent deviation values (residuals) for all calibration levels ≤15% from their respective theoretical values using a linear calibration model. The LOQ was 3 μg/g for liquid and 0.3 μg/collection (corresponding to 1 μg/g of consumed e-liquid). Furthermore, the aerosol collection trapping efficiency study indicated that over 99% of formaldehyde was collected with one pad and one impinger, while formaldehyde was not observed in the 2nd impinger.

The yield of acetal adducts in aerosol emissions was determined by GC-MS. The GC–MS system consisted of an Agilent 7980 gas chromatograph system coupled with a 5977A MS single quadrupole mass analyzer, equipped with a conventional electron ionization (EI) source. The chromatographic separation was conducted on a Rtx®-624 fused-silica capillary column (30 m × 0.25 mm × 1.4 μm film thickness) crossbonded with (6% cyanopropylphenyl/49% dimethylpolysiloxane phase), purchased from Restek (Bellefonte, PA, USA). An optimized GC oven temperature program was established where the oven temperature was initially held at 50 °C for 2 min, ramped to 75 °C at a rate of 10 °C/min, and then ramped to 235 °C at 10 °C/min and held for 3 min. Helium was used as the carrier gas at a constant flow rate of 1.4 mL/min. The GC injector was set to 230 °C, and 2 μL aliquots of samples were injected in splitless mode. The EI mass spectra for Gly-A, PG-A, d6-benzene, and 2,3-hexandione were acquired in EI mode (−70 eV) by monitoring their respective [M$^+$] molecular species (*m/z* 104, *m/z* 88, *m/z* 84, and *m/z* 114, respectively) with the dwell time value set at 50 milliseconds. The ion source and quadrupole temperatures were set at 230 °C and 150 °C, respectively. The concentrations of PG-A and Gly-A adducts in calibration standard solutions ranged between 0.01 and 2 μg/mL. The LOQ was determined as 0.8 μg/g of e-liquid and 2 μg/collection (corresponding to 0.8 μg/g of consumed e-liquid).

**Analytical experiments.** The investigatory approach and analytical experiments undertaken in this study are summarized in Figure 3. We first examined the behavior of formaldehyde-containing adducts in the acidic DNPH solution to verify the factual significance of the theory asserted by Jensen (Jensen et al., 2015), suggesting a pseudo-irreversible conversion of hemiacetal to acetal (1,1-geminal diether) induced by a unidirectional shift in hemiacetal/acetal equilibrium. The latter phenomenon occurs, according to Jensen et al., under a synergic effect arisen from the low-pH environment and the high abundance of PG and Gly (containing 2 and 3 hydroxyl moieties, respectively) in the reaction condition.

Little is known with respect to the formation of acetals in aerosol. Additional puffing experiments were also conducted to verify the possibility of the formation of formaldehyde-acetal adducts during the aerosolization process.

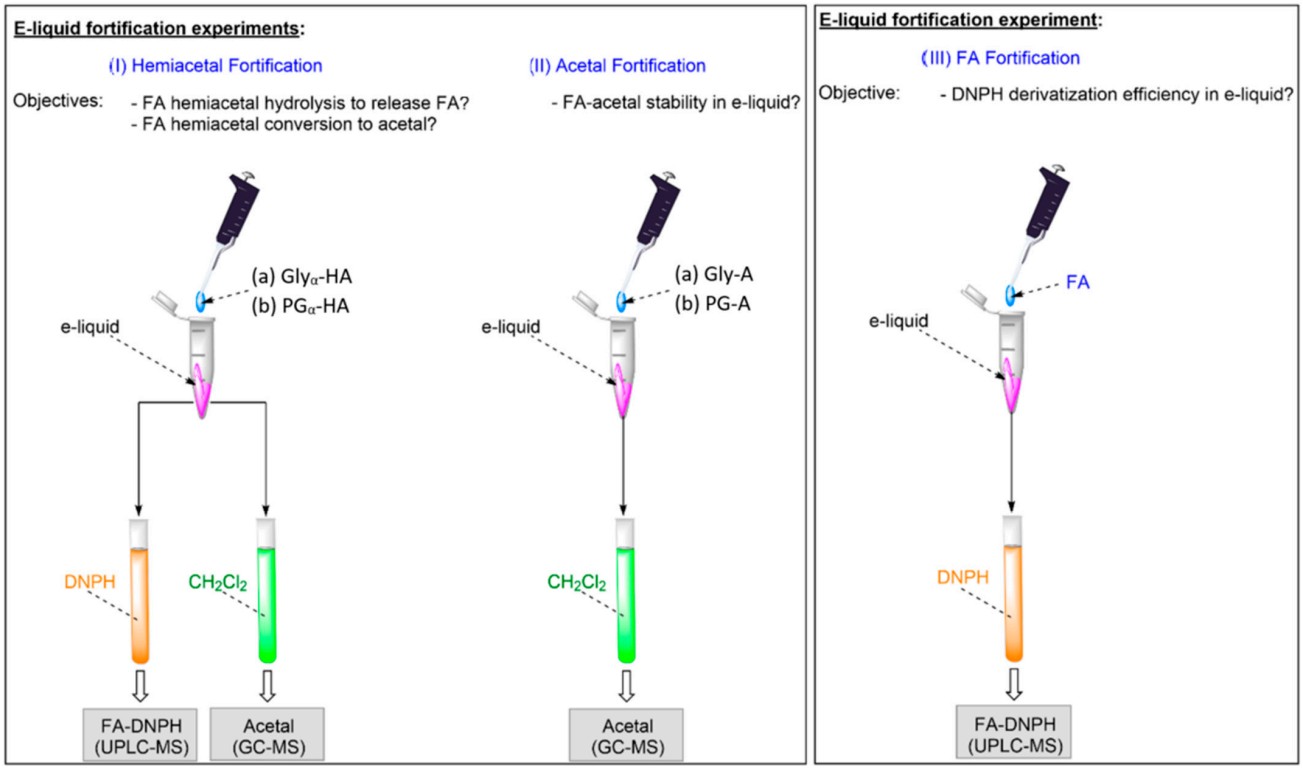

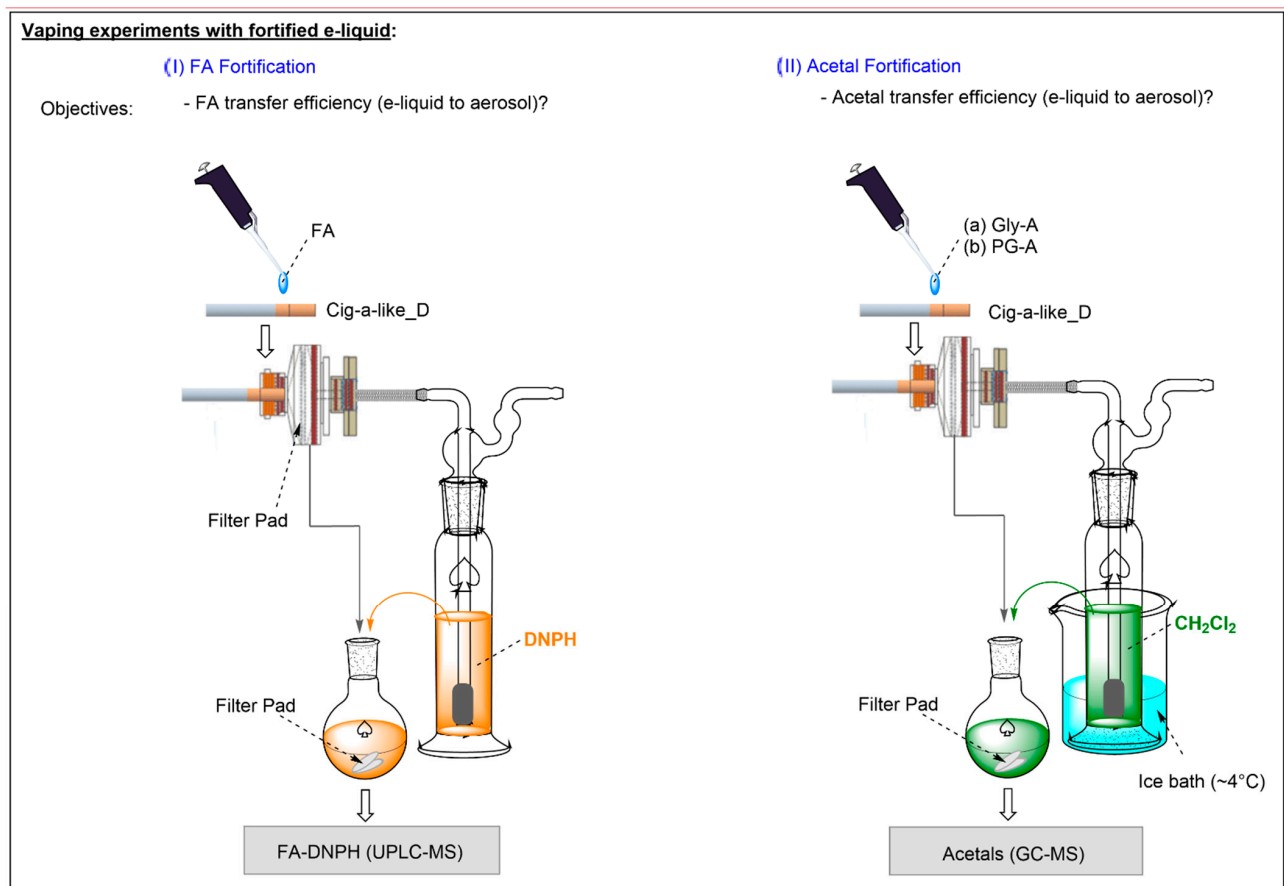

**Figure 3.** Summary of investigatory approach and analytical experiments.

## 3. Results and Discussion

### 3.1. Hemiacetal Behavior in Acidic DNPH Environment

#### 3.1.1. Investigating Potential Intramolecular Cyclization for FA-Hemiacetal to Acetals

To determine whether hemiacetal adducts undergo hydrolysis in the acidic DNPH solution, 20 mg of PGα-HA and Glyα-HA was added into two separate 20 mL aliquots of DNPH derivatization solution. The fortified mixtures were shaken for 30 s and further diluted with additional DNPH solution, resulting in hemiacetal concentrations of 5.86 μg/mL (Glyα-HA) and 3.19 μg/mL (PGα-HA). The fortified mixtures were then treated according to the procedure described earlier for e-liquid samples (Materials and Methods section), and their FA-acetal adducts (PG-A and Gly-A) were quantified by GC–MS, as described in the Analytical Method subsection. The formation of acetal adducts was deemed "confirmed" by comparing the retention times and mass spectral data to the corresponding commercially available material.

Figure 4 illustrates chromatographic traces for acetal molecular species acquired in fortified mixtures and acetal standard solutions (approximately 5 μg/mL). No acetal adducts were detected in fortified samples, indicating that formaldehyde-hemiacetal adducts did not convert to their respective acetals in the studied reaction environment (i.e., acidic DNPH). These results contradict Jensen's theory [36] of the unidirectional shift in hemiacetal/acetal equilibrium to form acetal adducts in the acidic DNPH environment.

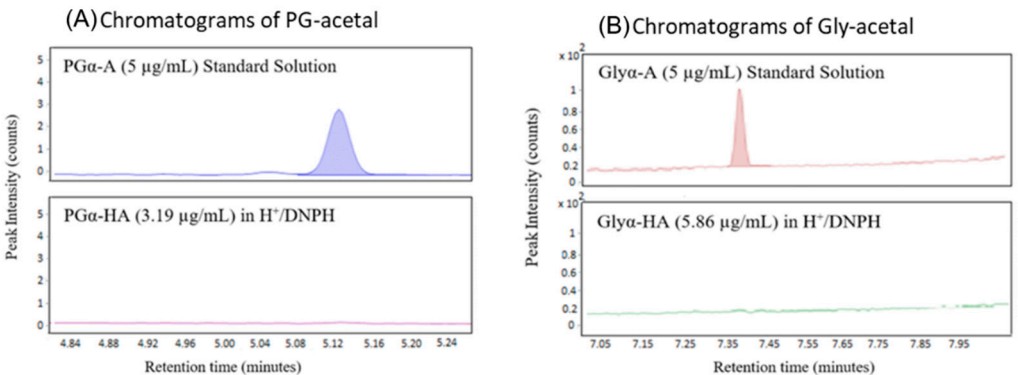

**Figure 4.** Comparison of chromatographic traces for PG-A (**A**) and Gly-A (**B**): DNPH fortified with PGα-HA and Glyα-HA (bottom) vs. standard solutions of FA-acetal adducts (top). FA-acetal adducts were analyzed by GC–MS, as described in the Analytical Method subsection.

#### 3.1.2. Investigating Potential Hydrolysis of FA-Hemiacetal Adducts to Release Formaldehyde

Additional experiments were conducted to verify whether FA-containing hemiacetals can undergo hydrolysis to release FA in the acidic DNPH solution (Scheme 2). Known amounts of PGα-HA and Glyα-HA were added to the DNPH derivatization solution to yield concentrations at 3.19 μg/mL (PGα-HA) and 5.86 μg/mL (Glyα-HA). In the event FA-hemiacetal hydrolysis occurs, the released FA is assumed to be readily derivatized to generate FA-DNPH-hydrazone, which is quantifiable by UPLC–MS. The theoretical (expected) and measured formaldehyde concentrations in the acidic DNPH are reported in Table 2.

**Table 2.** Average (n = 3) conversion percentage of (Glyα-HA and PGα-HA) adducts to formaldehyde in acidic DNPH environment.

| FA-Hemiacetal Adduct | [FA] Expected (μg/mL) | Average (n = 3) [FA] Measured (μg/mL) | % Hydrolysis of FA-HA Adducts in H$^+$/DNPH |
|---|---|---|---|
| PGα-HA | 0.93 | 0.96 | 103 |
| Glyα-HA | 1.44 | 1.50 | 104 |

**Scheme 2.** FA-hemiacetal hydrolysis (colors are used to visualize the distribution of oxygen atoms between the reaction products).

The conversion rates (% hydrolysis) of 103–104% reported in Table 2 demonstrate that both hemiacetal adducts, i.e., Gly$\alpha$-HA and PG$\alpha$-HA, undergo complete hydrolysis in the acidic environment and release formaldehyde. The latter readily reacts with DNPH reagent (present in large excess) to form the corresponding hydrazone (FA-DNPH). These results are in agreement with Knorr's report that the FA-DNPH derivative yields measured in the aerosol extract cover both free formaldehyde, as well as formaldehyde from PG$\alpha$-and Gly$\alpha$-HA that may be present in the solution [42].

*3.2. Acetal Reactivity and Formation Experiments*

3.2.1. Investigating Hydrolysis of Cyclic Formaldehyde-Acetal Adducts (Gly-A and PG-A) Acidic DNPH Environment

The release of formaldehyde from cyclic Gly-A and PG-A adducts requires two consecutive acid-catalyzed hydrolytic reactions involving the formation of an intermediate hemiacetal (Gly-HA and PG-HA, respectively). This hypothesis was investigated by fortifying a reference e-liquid formula (15% water, 2.5% NBW in a 50/50 mixture of PG and Gly) with known amounts of PG-A and Gly-A adducts. The formation of the intermediate hemiacetal adducts in H$^+$/DNPH was investigated by measuring the FA-DNPH hydrazone formed between FA (released by complete hydrolysis of hemiacetal) and the DNPH reagent (UPLC–MS analysis). FA-DNPH hydrazone was not detected in the DNPH extract solution, suggesting that the intermediate FA-hemiacetal was either not formed or formed and readily released FA in the H$^+$/DNPH environment. This finding allows us to demonstrate that the hydrolysis of cyclic FA-acetal does not occur in the acidified DNPH environment.

3.2.2. Evaluation of Formaldehyde-Acetals (Gly-A and PG-A) Formation as a By-Product of the Aerosolization Process

Puffing experiments were conducted on both cig-a-like and pod-type products listed in Table 1, to verify the possibility of the formation of formaldehyde-acetal adducts (Gly-A and PG-A) during the aerosolization process. Immediately after aerosol collection, the filter pad was extracted in the impinger solution, and acetal adducts were quantified by GC–MS. Figures 5 and 6 summarize averaged acetal yields measured in e-liquids and aerosols for each product.

**Gly-A Adduct**: Figure 5 shows that, except for the cig-a-like CCE4 exhibiting relatively high Gly-A levels in both e-liquid and aerosol (~70 µg/g), the Gly-A levels in all other cig-a-like e-liquids and aerosol emissions ranged between 0.5 and 6 µg/g. With respect to cig-a-like products, we recorded an excellent correlation (linear regression, R$^2$ = 0.999) between Gly-A content in the e-liquids vs. its yield in the corresponding aerosol (Figure 5, inset plot). The Gly-A levels in aerosol were similar to those in the corresponding e-liquid. This observation led us to conclude that the presence of Gly-A in the cig-a-like device aerosol occurs predominantly from the transfer of the adduct from e-liquids into the corresponding aerosols, as opposed to the adduct being formed by an acetalization reaction taking place during the aerosolization process.

With respect to pod category products (Figure 6), one of the three devices (i.e., PGE11) showed a significant increase in Gly-A yield in aerosol (over 2.5 fold increase as compared to the e-liquid), indicating that Gly-A is also formed as an aerosolization by-product for this specific pod product.

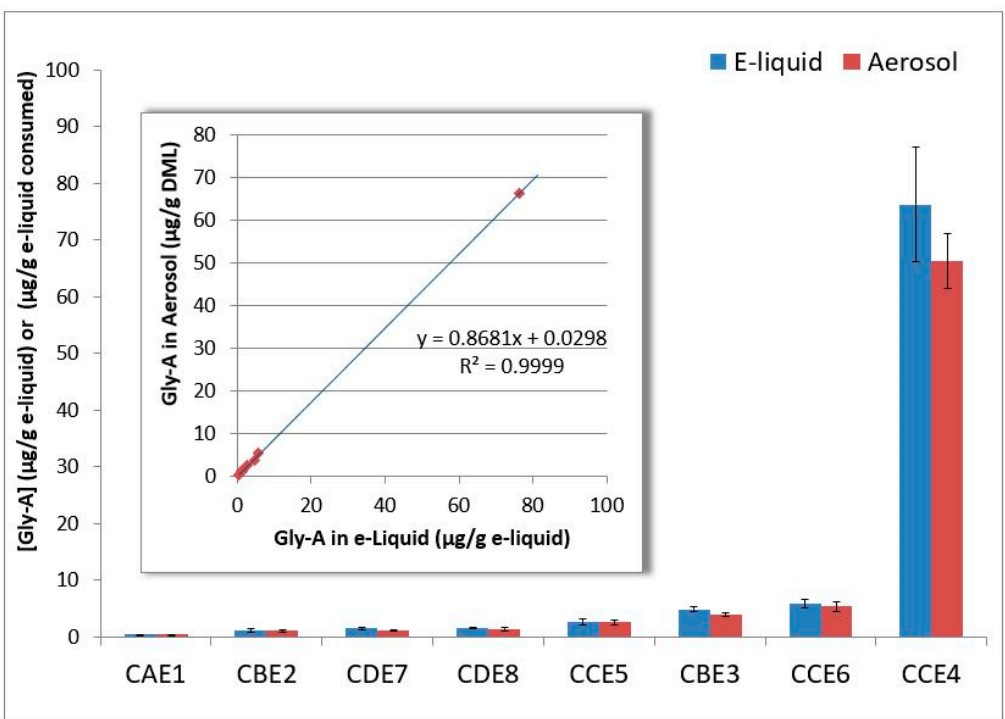

**Figure 5.** Gly-A average concentrations (n = 4) measured in e-liquid (μg/g e-liquid) and aerosol emission (μg/g e-liquid consumed) cig-a-like device category. Inset plot: e-liquid content vs. aerosol emissions.

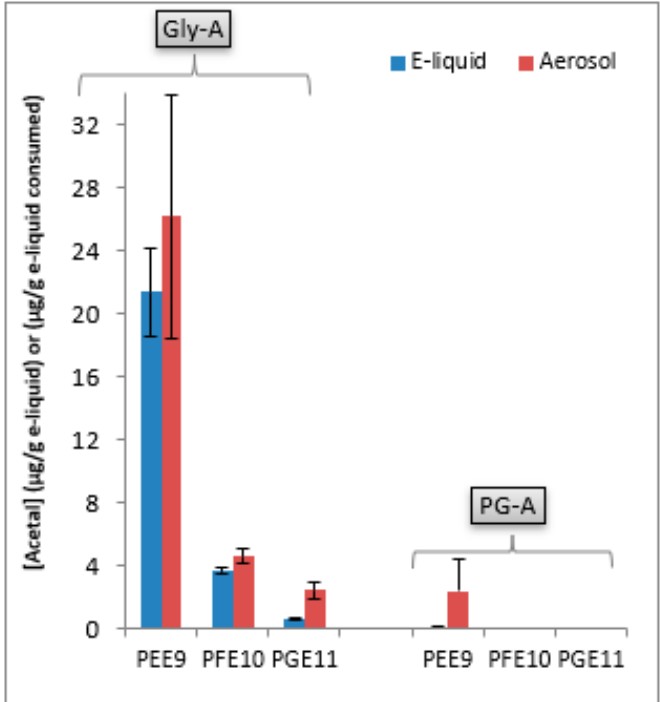

**Figure 6.** Gly-A and PG-A average concentrations (n = 4) measured in e-liquid (μg/g of e-liquid) and e-cigarette aerosol (μg/g of e-liquid consumed) Pod device category: Gly-A significantly increased in PGE11 aerosol while PG-A was only detected in PEE9. Error bars represent the standard deviation of each dataset.

**PG-A Adduct:** The PG-A was not detected in e-liquid or aerosol emission of any of the cig-a-like products investigated. With respect to pod category products, low μg/g levels

were detected in one of the three tested pod products, i.e., PEE9 (Figure 6). The formation of PG-A in E9 e-liquid might be due to an acetalization reaction occurring in the e-liquid that may have flavor added. A recent study published in Nicotine & Tobacco Research [43] reported that acetalization reactions could occur between PG hydroxyl moieties and flavor aldehydes (i.e., benzaldehyde, cinnamaldehyde, citral, ethylvanillin, and vanillin) to form aldehyde acetals in chemically reactive e-liquids. With respect to puffing experiments, the PG-A yield in aerosols generated from the PE device exhibited a significant increase (−18 times), compared to its measured content in the E9-flavored e-liquid (Figure 6). The increase in the amount of PG-A adduct in the aerosol may be attributed to an acetalization reaction taking place during the aerosolization process of the E9-flavored e-liquid in the PE device. To evaluate and compare the reciprocal influences of e-liquid composition and e-cig design (emission profile) on the acetalization reaction, a series of acetal-fortification experiments were conducted using an unflavored reference e-liquid that are discussed in the next section.

### 3.3. Investigating Formation of PG-Acetal in PEE9 Aerosol: Formation during Aerosolization vs. Transfer from e-Liquid to Aerosol

To confirm the hypotheses put forward with regard to the acetal formation pathway (i.e., formation during aerosolization or transfer from e-liquid to aerosol), a series of fortification experiments were conducted in which known amounts of PG- and Gly-acetal adducts were fortified (separate experiments) into the reference formulation (15% water, 2.5% NBW in a 50/50 mixture of PG and Gly). The fortified e-liquids were loaded into empty PE cartridges. Aerosol collection and analytical procedures used for the quantification of acetal adducts in fortified e-liquids and their yields in aerosol emissions are described in Figure 3. The fortification amounts added to the unflavored reference e-liquid were such as to ensure that acetal levels (if formed) are above the method limit of quantification (0.8 µg/g of e-liquid or e-liquid consumed).

The results of quantitative analysis for acetal adducts (Gly-A and PG-A) are summarized and presented as the average of four replicate observations (Figure 7). To evaluate and compare acetal levels in the e-liquid and aerosol, their detected quantities are expressed in µg/g of e-liquid and µg/g of vaporized e-liquid, respectively. The Gly-A concentrations in e-liquid and aerosol phases are not statistically different. Conversely, PG-A concentration was augmented, on average, from 5.9 (e-liquid) to 9.3 (aerosol) µg/g, corresponding to an approximately 60% increase in PG-A during aerosolization. The increased amount of PG-A adduct in unflavored e-liquid aerosol (+60%) is markedly lower as compared to the increase (18 times) observed for the E9-flavored e-liquid aerosol (Figure 6). This observation led us to conclude that the formation of the PG acetal adduct (PG-A) in aerosol is predominantly driven by the flavor composition in the e-liquid E9 as opposed to the design of the PE device.

### 3.4. Evaluation of the Efficiency of DNPH Derivatization Method

To investigate the method accuracy for the quantification of FA in e-liquid and aerosol, two commercially available CDE7 (nonflavored) and CDE8 (flavored) cartridges were used. Prior to puffing, e-liquid contents (E7 and E8) were removed from 11 cartridges, combined for each sample type, aliquoted, and then fortified with known amounts of formaldehyde. The formaldehyde concentration in fortified e-liquid was at −20 µg/g of e-liquid. The unfortified (background level of formaldehyde in the matrix) and fortified e-liquid samples were loaded into empty CD cartridges. Aerosol collection and analyses of formaldehyde levels in e-liquid (prior to puffing) and e-cigarette aerosol were conducted as described earlier. Table 3 shows the method accuracy calculated from these fortification experiments when the method was applied to e-liquid and aerosol. The excellent formaldehyde recovery values of 97.1–105.5% reported in Table 3 indicate that the formaldehyde derivatization by DNPH is the predominant reaction under study conditions; therefore, the method is fit for the quantification of formaldehyde in e-cigarette products.

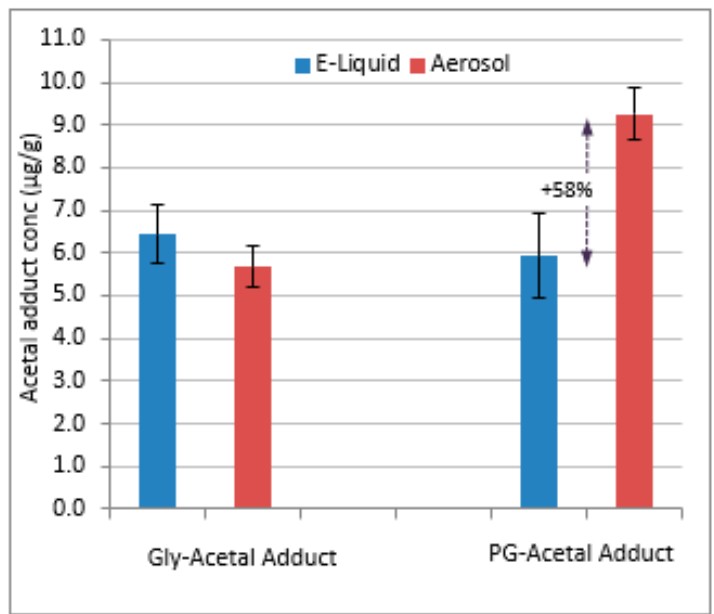

**Figure 7.** Comparison of average amounts (n = 4) of acetal adducts measured in fortified reference e-liquid (15% water, 2.5% NBW in a 50/50 mixture of PG and Gly) and aerosol emission (CD device).

**Table 3.** Method accuracy (fortification experiments, n = 4): %recovery values of formaldehyde in e-liquid and aerosol emission (CDE7 and CDE8) of 50 puffs.

| e-Liquid | | Unfortified Sample Concentration (μg/g) | Fortified Sample Concentration (μg/g) | Fortified Concentration (μg/g) | %Recovery (%) |
|---|---|---|---|---|---|
| CDE 7 | Average | 2.72 | 23.50 | 19.78 | 105.1 |
| | SD | 0.032 | 2.13 | | 10.8 |
| | %RSD | 1.2 | 9.1 | | 10 |
| CDE 8 | Average | 14.35 | 33.90 | 19.92 | 98.1 |
| | SD | 0.12 | 0.27 | | 1.3 |
| | %RSD | 0.84 | 0.80 | | 1.4 |

| Aerosol | | Unfortified Sample Concentration (μg/g e-Liquid Consumed) | Fortified Sample Concentration (μg/g e-Liquid Consumed) | Fortified Concentration (μg/g) | %Recovery (%) |
|---|---|---|---|---|---|
| CDE 7 | Average | 19.14 | 38.45 | 19.78 | 97.6 |
| | SD | 2.07 | 2.17 | | 11.0 |
| | %RSD | 11 | 5.6 | | 11.2 |
| CDE 8 | Average | 22.39 | 41.75 | 19.92 | 97.1 |
| | SD | 0.67 | 0.51 | | 2.6 |
| | %RSD | 3.0 | 1.2 | | 2.6 |

All concurring reactions/equilibria between participating reactants in the acidified environment are summarized in Figure 8. DNPH (0.6 mmoles) is in large excess as compared to the formaldehyde (0.04 mmoles, assuming an averaged FA emission rate of 25 μg/puff, 50 puffs collected). The reaction media is in a state of equilibrium governed by Le Chatelier's principle. The latter stipulates that a system in a state of equilibrium counteracts any perturbation by reaching a new equilibrium state. The consumption of formaldehyde by DNPH in the media is readily compensated by a shift in formaldehyde-hemiacetal hydrolysis (to release formaldehyde).

**Figure 8.** Concurring chain reactions between participating reactants in H+/DNPH environment.

## 4. Conclusions

The goal of this study was to provide an objective assessment of the DNPH method performance with respect to formaldehyde quantification in e-cigarette emissions. Our findings are in contradiction with a publication by the Jensen group (Jensen et al., 2015), which suggested that formaldehyde levels in ENDS products were underreported because formaldehyde may react with e-liquid excipients (PG and Gly) in the aerosol to form hemiacetals, which, in turn, form cyclic acetals in the acidic DNPH trapping solution.

The results from our investigations, focused on the behavior of formaldehyde-containing hemiacetal adducts in the acidic DNPH solution, clearly demonstrated that these compounds undergo a complete hydrolysis in the acidic environment to release formaldehyde, which is then derivatized by DNPH to form formaldehyde-hydrazone (FA-DNPH). Conversely, acetals of PG and Gly added to the DNPH trapping solution would not hydrolyze to form the hydrazone.

Our results from machine-generated aerosols showed that the aerosol generated from formaldehyde-fortified e-liquids provided quantitative recovery of formaldehyde in the aerosol, suggesting that if any hemiacetal was formed in the aerosol, it would readily hydrolyze to free formaldehyde in the acidic DNPH trapping solution. We believe that the presence of derivatization agent (DNPH) at a large excess in the acidic solution exerts a major role on hemiacetal/formaldehyde equilibrium: the hemiacetal/formaldehyde equilibrium shifts from the hemiacetal to the formaldehyde due to a complete and rapid consumption of free formaldehyde by DNPH.

We also demonstrated that acetal adducts fortified into e-liquids are almost completely transferred (−90%) to the aerosol during aerosolization in both device categories. Additionally, we observed that in the case of one of the tested pod devices (PE), the PG-acetal adduct can also be formed via an acetalization reaction during the aerosolization process.

We believe that our evidence-based analytic approach provides an objective assessment of DNPH method performance. The results of this study demonstrate that the measured FA-DNPH yields in the aerosol of the e-cigarettes account for all unreacted formaldehyde and formaldehyde–hemiacetal adducts and, therefore, the DNPH derivatization method allows for an accurate measurement of formaldehyde amounts in e-cigarette liquids and aerosols.

**Author Contributions:** Conceptualization, Y.B.P., K.A.W., W.P.G. and M.S.M.; formal analysis, X.C.J., R.M.B. and K.C.A.; investigation, M.S. and X.C.J.; methodology, M.S.M. and X.C.J.; project administration, X.C.J.; resources, R.M.B. and K.C.A.; supervision, K.A.W.; visualization, X.C.J. and M.S.; writing—original draft, M.S.; writing—review and editing, M.S., M.S.M., X.C.J., W.P.G. and Y.B.P. All authors have read and agreed to the published version of the manuscript.

**Funding:** This research received no external funding.

**Institutional Review Board Statement:** Not applicable.

**Informed Consent Statement:** Not applicable.

**Acknowledgments:** The authors would like to express their gratitude to Jennifer H. Smith, Jason W. Flora, Fadi Aldeek, Cynthia Cecil, and Robin Brownhill for their excellent comments and suggestions during review of this manuscript.

**Conflicts of Interest:** The authors declare no conflict of interest.

**Abbreviations**

CFP, Cambridge Filter Pad; CORESTA, Centre de Coopération pour les Recherches Scientifiques Relatives au Tabac (Centre for Scientific Research Relative to Tobacco); CRM, CORESTA recommended method; DNPH, 2,4-dinitrophenlhydrazine; ENDS, electronic nicotine delivery system; FA, formaldehyde; FA-DNPH, formaldehyde-DNPH hydrazone; FDA, U.S. Food and Drug Administration; GC–MS, gas chromatography–mass spectrometry; Gly, glycerin; Gly-A, glycerin-acetal adduct; Glyα-HA, glycerin-hemiacetal adduct; HPHC, harmful and potentially harmful constituent; HPLC-DAD, high-performance liquid chromatography-diode array detector; ICH, international conference on harmonization; ISO, international organization for standardization; LC–MS/MS, liquid chromatography–tandem mass spectrometry; NBW, nicotine by weight; PG, propylene glycol; PGα-HA, propylene glycol-hemiacetal adduct; PG-A, propylene glycol-acetal adduct; PFBHA, O-(2,3,4,5,6-pentafluorobenzyl)hydroxylamine; SIM, single ion monitoring; SPME, solid-phase microextraction; UPLC–MS/MS, ultra-performance liquid chromatography–tandem mass spectrometry; UV, ultraviolet; WHO, World Health Organization.

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
