# Peer review of "Determination of Formaldehyde Yields in E-Cigarette Aerosols: An Evaluation of the Efficiency of the DNPH Derivatization Method"

_separations, doi:10.3390/separations8090151_

Round 1

Reviewer 1 Report

The paper is well written, I have only minor comments:

  • Line 42: Font is different.
  • Hydrolysis rate presented in table 2 is based on how many experiments.
  • Figure 5: Why CBE3 is so different from other? It seems like a outlier, better not include or provide explanation.
  • Figure6: What these error bar means? How they have been calculated?

Author Response

  1. Line 42: Font is different: Corrected
  2. Hydrolysis rate presented in table 2 is based on how many experiments: The %hydrolysis values in the table are average results calculated for three independent  experiments. Table title was revised  to include of replicate observations (n=3).
  3. Figure 5: Why CBE3 is so different from other? It seems like a outlier, better not include or provide explanation: It is sample CCE4 (and not CBE3) exhibiting significantly higher levels.  The high results have been investigated/confirmed: high adduct levels in both e-liquid/aerosol are product specific. However, the inclusion of CCE4 data confirm that the adduct is transferred from e-liquids into the corresponding aerosols (as opposed to being the product of an acetalization reaction taking place during aerosolization process) independently of adduct level in e-liquid.
  4. Figure 6: What these error bar means? How they have been calculated? The caption for Figure 6 was revised: "Error bars represent the standard deviation of each data set."  

Reviewer 2 Report

The manuscript is well written and covers an important topic in ENDS research and Harm Reduction. 

Line 45.  Reference 10 on line 45 is unnecessary.

Line 47.  “Formaldehyde has also been reported in e-cigarette emissions”.  This statement needs current references.

Lines 51-55.  Needs references to support this statement.  Consider replacing “quality” with “design”.

Line 69:  “Due to its high reactivity (formaldehyde)”. Also due to its propensity to form paraformaldehyde, low molecular mass, and lack of a strong chromophore.   

Lines 80-83.  FA-DNPH is used in several standardized methods outside of nicotine products including US-EPA, NIOSH, and ISO.  Adding a sentence here on the broader use of DNPH would be helpful. 

Line 116.  “published in New England Journal of Medicine” is not needed and seems out of place.

Line 147.  “empty Cig-a-like E cartridges” This requires additional detail since empty devices are typically not sold are retail (see line 138-139).  Were these empty devices produced by cleaning full products or were the empty devices supported by the manufacture?  

Line 165-166.  (19mM DNPH) please add citation if this was adapted from a standardized method ISO, CORESTA etc.   

Line 177 “Drechsel-type bottle container”. Impinger design is a critical aspect of the method, please add supplier and part number if available.

Line 203.  This LC-MS method is similar to the authors’ prior work.  Journal of Chromatographic Science, 2017, Vol. 55, No. 2, 142–148.  If possible, please reference this prior work and remove duplicate information. 

Analytical methods

List the make model of the MS/MS. 

Provide the city and state with the first usage of the company name. Check entire section. 

The following items are typically included in an ICH validation report.  Accuracy, Precision, Repeatability, Intermediate Precision, Specificity, Detection Limit, Quantitation Limit, Linearity, and Range. Accuracy, Intermediate Precision, and Specificity are not addressed in the text and should be added. For linearity, list the deviation of the actual data points from the regression line.

The validation status of the GC-MS is not stated.  Please clarify if this method has been validated according to ICH guidance. 

Line 256.  Avoid the use of vaping when referring to device activation.  Puffing is preferred.  Also, line 323.  Puffing and vaping are both used in the text.

Line 341 Avoid the use of vaporization when referring to the formation of the aerosol.  Aerosolization is preferred.  Vaporization and aerosolization are both used in the text.

Line 391. Did the authors consider the impact of e-liquid pH on the formation of PG-A?  

Line 407.  “milieu” Consider a different work choice for clarity. 

Figure 3.  The device looks like a cigarette.  If possible, please use something without tipping paper or label the device as “e-cigarette” for clarity. 

Author Response

  1. Line 45.  Reference 10 on line 45 is unnecessary. Reference was deleted.
  2. Line 47.  “Formaldehyde has also been reported in e-cigarette emissions”: This statement needs current references. Additional references were added. 
  3. Lines 51-55.  Needs references to support this statement.  Consider replacing “quality” with “design”. The various factors/parameters affecting e-cigarette performance are explained (including citations/references ) in the subsequent section (lines 60-68). 
  4. Line 69:  “Due to its high reactivity (formaldehyde)”. Also due to its propensity to form paraformaldehyde, low molecular mass, and lack of a strong chromophore. : the suggested reasons were inserted.
  5. Lines 80-83.  FA-DNPH is used in several standardized methods outside of nicotine products including US-EPA, NIOSH, and ISO.  Adding a sentence here on the broader use of DNPH would be helpful.: Additional info added as suggested (Lines 72-75).
  6. Line 116.  “published in New England Journal of Medicine” is not needed and seems out of place: Removed “published in New England Journal of Medicine”
  7. Line 147.  “empty Cig-a-like E cartridges” This requires additional detail since empty devices are typically not sold are retail (see line 138-139).  Were these empty devices produced by cleaning full products or were the empty devices supported by the manufacture? : Additional text inserted. The cartridges were from commercial products that were emptied (e-liquid was removed) prior to re-filled. 
  8. Line 165-166.  (19mM DNPH) please add citation if this was adapted from a standardized method ISO, CORESTA etc. : CORESTA method reference was cited.  
  9. Line 177 “Drechsel-type bottle container”. Impinger design is a critical aspect of the method, please add supplier and part number if available: Info was inserted. 
  10. Line 203.  This LC-MS method is similar to the authors’ prior work.  Journal of Chromatographic Science, 2017, Vol. 55, No. 2, 142–148.  If possible, please reference this prior work and remove duplicate information.:  Although the method is based on the reference method, there are a lot of changes made to the method reference. Therefore, we believe it is important to maintain the detailed description.

Analytical methods

  1. List the make model of the MS/MS.: Info inserted as per suggestion.
  2. Provide the city and state with the first usage of the company name. Check entire section.: Info was added/updated.
  3. The following items are typically included in an ICH validation report.  Accuracy, Precision, Repeatability, Intermediate Precision, Specificity, Detection Limit, Quantitation Limit, Linearity, and Range. Accuracy, Intermediate Precision, and Specificity are not addressed in the text and should be added. For linearity, list the deviation of the actual data points from the regression line. : Info on method accuracy, intermediate precision and specificity was inserted. 
  4. The validation status of the GC-MS is not stated.  Please clarify if this method has been validated according to ICH guidance. : This method was partially validated and not according to ICH guidance. 
  5. Line 256.  Avoid the use of vaping when referring to device activation.  Puffing is preferred.  Also, line 323.  Puffing and vaping are both used in the text. : Changes made throughout the document as per suggestion.
  6. Line 341 Avoid the use of vaporization when referring to the formation of the aerosol.  Aerosolization is preferred.  Vaporization and aerosolization are both used in the text.: Changes were made throughout the document as suggested. 
  7. Line 391. Did the authors consider the impact of e-liquid pH on the formation of PG-A?: For this study, we focused on the adduct behavior in acidic DNPH environment. We have not data to support the impact of pH. 
  8. Line 407.  “milieu” Consider a different work choice for clarity.:  Changes were made (milieu was replaced by "environment" or "condition" throughout the document (line 259, Line 281 and 411). 
  9. Figure 3.  The device looks like a cigarette.  If possible, please use something without tipping paper or label the device as “e-cigarette” for clarity.: Figure 3 was updated by specifying the e-cigarette type (i.e. Cig-a-like).